# Multi-Domain Rapid Enhancement Networks for Underwater Images

**DOI:** 10.3390/s23218983

**Published:** 2023-11-05

**Authors:** Longgang Zhao, Seok-Won Lee

**Affiliations:** 1The Knowledge-Intensive Software Engineering (NiSE) Research Group, Department of Artificial Intelligence, Ajou University, Suwon City 16499, Republic of Korea; zhaolonggang@ajou.ac.kr; 2Department of Software and Computer Engineering, Ajou University, Suwon City 16499, Republic of Korea

**Keywords:** underwater image enhancement, multi-domain machine learning, DCNN, domain adaptability, perceptual loss

## Abstract

Images captured during marine engineering operations suffer from color distortion and low contrast. Underwater image enhancement helps to alleviate these problems. Many deep learning models can infer multi-source data, where images with different perspectives exist from multiple sources. To this end, we propose a multichannel deep convolutional neural network (MDCNN) linked to a VGG that can target multi-source (multi-domain) underwater image enhancement. The designed MDCNN feeds data from different domains into separate channels and implements parameters by linking VGGs, which improves the domain adaptation of the model. In addition, to optimize performance, multi-domain image perception loss functions, multilabel soft edge loss for specific image enhancement tasks, pixel-level loss, and external monitoring loss for edge sharpness preprocessing are proposed. These loss functions are set to effectively enhance the structural and textural similarity of underwater images. A series of qualitative and quantitative experiments demonstrate that our model is superior to the state-of-the-art Shallow UWnet in terms of UIQM, and the performance evaluation conducted on different datasets increased by 0.11 on average.

## 1. Introduction

With the increasing demand for ocean cognition and situational awareness, exploring the mystery of ocean depths and obtaining high-quality underwater images are urgently needed by many marine enterprises [1,2]. Maritime enterprises rely on thousands of Internet of Things (IoT) sensors scattered underwater to collect data on the activity status of underwater creatures and moving images of seabed plates. This information is used to improve awareness of the continuous situation of the ocean. Owing to breakthroughs in key technologies such as big data, cloud computing, and the IoT, maritime enterprise information management can realize a clear perception of the entire underwater domain, providing important support for ocean exploration, national defense, and security [3].

In recent years, to solve engineering problems such as marine environmental monitoring, submarine surveying and mapping, submarine archaeological exploration, garbage collection, and underwater rescue, nations worldwide have vigorously developed underwater detection applications [4]. These applications require the real-time interpretation of images/videos so that underwater robots based on vision guidance can intelligently perceive the environment and set future execution tasks [5]. For example, Ref. [6] proposed an underwater image enhancement network named Ucolor, which uses medium transmission guidance, multicolor space embedding, and a combination of physical models and learning methods to solve the color deviation and low-contrast problems of underwater images. The authors of [7] proposed an underwater image enhancement method based on generative adversarial networks using a multiscale generator to generate clear underwater images, effectively correcting color casts and contrast problems while protecting detailed information. In, [8] an L2UWE framework is proposed to efficiently enhance low-light underwater images by relying on local contrast and multiscale fusion technology to improve the clarity and brightness of the image. The authors of [9] constructed an underwater image enhancement benchmark (UIEB) and proposed the WaterNet underwater image enhancement network that can effectively correct color casts and restore image details. The authors of [10] proposed a generative adversarial network based on Pix2Pix and introduced technologies such as deep residual learning and multilayer perceptrons to remove the fog effect, correct color shift, and increase image details. Although these methods have made significant progress in improving the color cast, contrast, and brightness of underwater images, they fail to fully consider the relationship between the contrast, brightness, and color of underwater images and fail to adaptively balance these factors. Future research should continue to address this issue to improve the effectiveness of underwater image enhancement.

Contrast deterioration and color distortion in underwater images limit the application of underwater vision tasks [11]. In addition, the wavelength decays exponentially in deep water, resulting in the absorption of red wavelengths, leading to the dominance of green or blue hues in underwater images. These problems severely degrade the visual effects of underwater images. Consequently, the same scene in underwater images presents different background lights, forming a multi-domain problem [12]. For example, a dimly lit image is considered to be from the same domain and a well-lit image is considered to be from another domain.

Underwater images from different viewpoints or light backgrounds, as shown in Figure 1, can be regarded as data from different domains. Multi-domain machine learning shares model parameters through the model training of data in different domains to improve the model learning efficiency, which enables the enhancement of models for underwater images in different domains. However, most multi-domain machine learning methods assume the same distribution of data in different domains, ignoring non-IIDs, and such neglect leads to the inability of the model to make its image enhancement ability the strongest in each domain.

Scholars have proposed a variety of underwater image-enhancing methods, such as nonphysical and physical model-based methods [13]. The nonphysical model improves image quality by adjusting the image pixel value without relying on the underwater imaging model. However, nonphysical and physical models ignore the attenuation characteristics of underwater images in different domains and cannot achieve multi-domain underwater image enhancement. Data-driven methods use deep learning to learn nonlinear feature maps and enhance the underwater images. Convolutional neural networks (CNNs), which are widely used in computer vision tasks, are invariant to displacement and distortion, and have good recognition capabilities [14]. Many models based on CNNs (e.g., Shallow UWnet [15] and Uresnet [16]) and generative adversarial networks (GANs) (e.g., FUnIE GAN [17], Water Net [18], and Cycle-GAN [19]) have been used to enhance image quality by learning from a large amount of data. Significant progress has been made in image super-resolution, denoising, deblurring, and dehazing techniques. However, deep learning models have a weak inference ability for solving multi-domain problems because they cannot effectively use the similarity of images between domains and ignore the local information of images in different domains, for example, the luminosity change in the same target [20].

This paper proposes a multi-domain underwater image fast enhancement model based on a multichannel deep convolutional neural network (DCNN) combined with a VGG network to improve the inference ability of multi-domain underwater image models called the MDCNN-VGG. The MDCNN-VGG has more layers and a more complex structure than the CNN. The DCNN can avoid the performance degradation of the network model caused by the data heterogeneity of underwater images in multiple domains. It uses DCNNs of different channels to mine the texture and color of underwater images in different domains, and this information is fed into the VGG. The VGG recognizes and classifies data elements in multiple domains to obtain specific feature representations in each domain, thereby achieving the rapid enhancement of underwater images.

The contribution of this study is as follows:

We designed a multi-domain underwater image enhancement model with a multichannel DCNN linked to a VGG; specifically, the different network streams designed in the DCNN shared parameters through back-and-forth passing to enhance domain adaptation. The importance of different model parameters is also selected in the soft mask configuration model, such that important model parameters (e.g., texture structure and color) are input to the VGG, which in turn yields a specific feature representation in each domain to enhance underwater images.

To optimize the performance of the MDCNN-VGG, we designed a perceptual loss function for multi-domain underwater image enhancement. Multilabel soft-margin loss is used for specific tasks, and VGG perceptual loss is used for external supervision to achieve pixel-level loss and for preprocessing edge sharpness, thereby enhancing the structure and texture similarity of underwater images. In turn, we can optimally adjust the coefficients in the perceptual loss function to control the involvement of different functional loss terms in the model-training process to detect the focal region of the input image for target class enhancement.

Qualitative and quantitative experiments showed that the enhancement effect of this model on underwater image quality was better than that of the benchmark model.

## 2. Literature Review

### 2.1. Deep Learning

Image enhancement is a topic of significant interest within the fields of computer vision, signal processing, and others. Earlier works used artificially created filters to enhance local colors for contrast/brightness improvement and global enhancement based on scene assumptions (e.g., fog lines, dark channels, etc.). With the development of deep learning and driven by large-scale datasets, image enhancement has been remarkably successful. For example, deep CNN-based models have achieved superior performance in solving image coloring, color/contrast adjustment, and de-cluttering [21]. The recently proposed Shallow UWnet is a gated fusion CNN trained on the UIEB dataset for underwater image enhancement [22]. UResnet, a CNN-based residual network, is proposed in [23] as a more comprehensive supervised learning method for underwater image enhancement. In addition, GAN-based image style transformation or generation has achieved great success. For example, the recently proposed FUnIE GAN is assumed to perform nonlinear mapping between distorted and enhanced images, and it removes image blurring by constructing an image enhancement problem [24]. The conditional GAN proposed in [25] learns image enhancement through generalized training on large-scale datasets. In contrast, bidirectional GANs (e.g., CycleGAN, DualGAN [26], etc.) address the practical application of image enhancement by using a cyclic consistency loss. However, the above studies do not take full advantage of the similarity of multi-domain images taken from different angles of the same target.

### 2.2. Physical-Based Methods 

Traditional physics-based methods use atmospheric scattering models to estimate light transmission and ambient temperature scenes to recover the true pixel intensity [27,28]. The underwater image-defogging algorithm proposed in [29] reduces information loss in the output image [30]. In addition, multiband fusion-based enhancement [31] and blurred line-based color recovery (Uw-HL [32]) have been used to recover underwater image quality.

### 2.3. Nonphysical-Based Methods 

Nonphysical-based methods directly modify image pixel values to produce satisfactory results without using a physical degradation model. In citeancuti2012 enhancing, a fusion-based method was proposed that can improve contrast and visual effects; however, an over-enhancement phenomenon exists. Recently, Ref. [33] modified [34] by introducing a novel white-balancing method to reduce over- and under-enhancement. Another study is based on the retinex model, [35] and uses color correction, post-enhancement, and layer decomposition to improve the underwater image quality.

In summary, deep learning techniques have achieved state-of-the-art performances in image enhancement tasks and can automatically learn relevant features from large datasets, thereby reducing the need for handcrafted features. However, deep learning models often require substantial computational resources and large training datasets. Overfitting can be a concern if the training data are not representative of the target domain [36].

Physics-based methods provide a solid theoretical foundation for image enhancement, allowing for accurate modeling of physical degradation. Physics-based methods are sensitive to the accuracy of assumed physical models, and deviations from these models can lead to errors.

Nonphysical-based methods are versatile and do not rely on explicit physical models, making them more flexible in a wider range of scenarios. An overreliance on nonphysical methods may lead to image artifacts or unrealistic enhancements.

## 3. MDCNN-VGG Hybrid Model Architecture 

### 3.1. Overall

In this study, the MDCNN-VGG is proposed to make full use of the differential distribution of information in different domains while combining it with its parameter-sharing mechanism to enhance domain adaptability [29]. Extensive qualitative and quantitative experimental results demonstrate that the MDCNN-VGG has better underwater image quality enhancement than the benchmark model.

As shown in Figure 2, the proposed MDCNN-VGG consists of a multichannel DCNN with a VGG-16 model that combines the advantages of neural network classifiers and VGG [33] perceived loss.

The MDCNN consists of multiple DCNNs in parallel, and the specific structure of each DCNN, which consists of multiple fully connected CNN layers, is shown in Figure 2. DCNNs are applied using the same principles as traditional CNNs, which employ alternating convolutional layers and pooling in their network structure with fully connected network ends. The most distinguishable features are extracted from the original input images using supervised learning. The effective subregions are computed from the original underwater images of different domains using the perceptual field features of the DCNN [37]. To enhance the model domain adaptation capability, two DCNN network streams are used to share parameters between them, and the importance of different parameters of the model is configured using a soft mask to enhance the information of the network stream. Information such as the texture, structure, and color of the underwater images from different domains are mined and fed into the subsequent VGG. We set different CNN channels so that in order to better extract underwater images in different domains, we map different channels to underwater images in different domains, and then perform feature fusion after different channels, so that the model can better obtain specially issued useful images, as well as better differentiation of different areas of underwater images.

The VGG-16 identifies and classifies data elements belonging to different underwater background categories to obtain a feature representation of an underwater image for each domain. Specifically, the MDCNN-VGG identifies and classifies data elements belonging to different underwater background categories (e.g., water bodies appearing blue–green or dark blue, rocks underwater appearing silver) based on VGG perceptual loss for each domain.

### 3.2. Single-Channel DCNN 

The architecture of the single-channel DCNN model is illustrated in Figure 3. The network model contains two network streams: $S_{cl}$ and $S_{com}$. $S_{cl}$ was developed with the goal of searching for regions that contribute to the identification of target objects in underwater images, and $S_{com}$ ensures that all regions favorable for identification are found.

The network stream $S_{cl}$ contains classical optimization techniques such as pooling, dropout, and other settings. The CNN is immediately followed by three fully connected layers. The parameters of this fully connected layer are shared with those of the fully connected tail layer. Immediately thereafter, the soft mask feature (see attention mechanism [35]) is used to configure the importance of the different model parameters, allowing attention operations to be performed on different domain underwater images in the DCNN for the task of interest. In other words, the soft mask of the network is trained in an end-to-end manner to achieve precise enhancement of the attention content (e.g., the specific color enhancement of the corresponding target), and the soft mask has an enhancing effect on the information of the previous network stream, whose activation function uses Mish [38]. The results are fed into the subsequent network stream based on the idea of Resnet [39] that links the two network streams before and after, thereby avoiding gradient disappearance and sharing the model parameters of the CNN of the preceding network stream $S_{cl}$ into the CNN of the following network stream. The specific implementation of parameter sharing in the model structure is shown by linking $S_{cl}$ and $S_{com}$ in Figure 3.

Specifically, in network stream $S_{cl}$, for a given domain of underwater images, I, $f_{l,k}$ is represented as the activation function of unit k in the lth layer of the soft mask. It can be seen that $f_{l,k}$ obtains the classification probability corresponding to domain category c for each underwater image to be enhanced, and the gradient obtained based on the activation function is used to update the weight, $\omega_{l,k}^{c}$, of the neurons through global average pooling, as shown in Equation (1).
(1)$$\omega_{l,k}^{c}=GAP\left(\frac{\partial \omega^{c}}{\partial f_{l,k}}\right)$$
where *GAP*(·) denotes the global average pooling operation; at this time, there is no need to use the backward pass method to obtain $\omega_{l,k}^{c}$. $\omega_{l,k}^{c}$ denotes the importance of $f_{l,k}$ support c-class underwater image enhancement in the soft mask. To enhance the generalization ability of the overall network model, the weight is used to represent the importance of c-class image probability, the 2D convolution operation is performed on all f_l_s to integrate the soft mask with the activation output of all layers, and then the Mish [38] operation is performed to obtain the soft mask ($A^{C}$)
(2)$$A^{C}=\mathrm{Mish}\left(\mathrm{conv}\left(f_{l},\omega^{c}\right)\right)$$

The soft mask applied to the original input underwater image is obtained through $A^{C}$ using Equation (3) to obtain $I^{*c}$, which represents the semantic information in the c-class underwater image on which the network model is currently focused.
(3)$$I^{*c}=I-\left(T\left(A^{c}\right)⊙I\right)$$
where $T\left(A^{c}\right)$ is a masking factor based on a threshold, and the sigmoid function is used as an approximation, as defined in Equation (4).
(4)$$T\left(A^{c}\right)=\frac{1}{1+\exp\left(-\omega \left(A^{c}-\sigma \right)\right)}$$
where ω is the scaling parameter. It is then used as an input to the network stream to obtain enhanced information regarding the different domains of the underwater image. The attention mechanism in the designed model guides the network to focus on all regions of interest; that is, the high-response region in the soft mask contains an image quality that can be enhanced. The loss function uses the pixel MSE to calculate the difference between images I and $I^{*c}$.
(5)$$L_{MSE}=\frac{1}{n}\sum_{c}s^{c}{\left(I_{i}-I_{i}^{*c}\right)}^{2}$$

To minimize the prediction probability error, we redesigned the MSE by adding $s^{c}\left(•\right)$ to the constraint, which denotes the prediction probability of class c, where *n* is the number of images, *I*.

Considering the need to share model parameters for multi-domain underwater images, we set $L_{\mathrm{self}}$ for the multi-domain underwater image enhancement objective function using multilabel soft marginal loss [40]. Alternative loss functions can be used for specific tasks to better separate individual domain categories. One of the simplest methods is to add margins to each domain, shown in Equation (6).
(6)$$L_{\mathrm{cl}}=-\frac{1}{\left|Q\right|}\sum_{(x,y)\in Q}\log\frac{e^{D\left(s^{c}\left(I_{i}-I_{i}^{*c}\right),r_{c}\right)}}{e^{D\left(s^{c}\left(I_{i}-I_{i}^{*c}\right),r_{c}\right)}+\sum_{k\in C}e^{D\left(s^{c}\left(I_{i}-I_{i}^{*c}\right),r_{k}\right)+m}}$$
where Q is the test set, and the representative points of each domain are represented by $r_{1},\ldots ,r_{c}$ (e.g., the centroids of all samples in the support set of each domain are used as representative points). D( ) is the metric module used to measure the cosine similarity of two feature vectors. The same margin, m, was added between two different domains, forcing a certain distance between the samples of different domains.

Subsequently, the overall loss function $L_{\mathrm{self}}$ of the model is the sum of the objective functions $L_{\mathrm{self}}$ and $L_{MSE}$ of the multi-domain underwater image deep learning, as defined in Equation (7).
(7)$$L_{\mathrm{self}}=L_{\mathrm{cl}}+\alpha L_{MSE}$$
where *α* is a weighting parameter, set empirically, and *α* = 1 is used for all experiments in this study. Guided by $L_{\mathrm{self}}$ updates to the model weights, the network model learns to extend as much as possible the focal region of the input image that contributes to the target class enhancement, thus allowing the soft mask to be tailored to the task of interest (i.e., underwater image enhancement). The model was trained using a multinomial loss function that considered the resulting pixel-level loss to preprocess the sharpness of the edges and enhance the structural and textural similarity of underwater images.

### 3.3. MDCNN-VGG 

In addition to allowing the network model to explore its own model weights, the network model itself can also employ additional supervised learning similar to the soft mask to make it suitable for the task of interest. We introduce the VGG [41] to integrate additional supervision into a supervised learning framework seamlessly.

There is a multi-domain phenomenon in underwater images in the application scenario studied in this paper; that is, there are multi-domain underwater images caused by factors such as different viewing angles or background light intensities. In our study, the multiple DCNNs described in Section 2.2 are used to integrate into multiple channels, and the advantages of the designed CNN model are described in Section 3.1. Based on the concept of ensemble learning [42], we set up multiple channels to learn the background types of underwater images in different domains. The model mines the effective information of underwater images of different domains from a local perspective with the help of DCNNs of different channels, while delivering this effective information to the VGG for accurate underwater image enhancement. The MDCNN-VGG effectively exploits the differential distribution of different domain information and applies a parameter-sharing-based mechanism to improve domain adaptation. In this study, the model learns the distribution bias of underwater image data from different domains to make it highly robust and to improve its generalization capability. The architecture of the designed MDCNN-VGG model is illustrated in Figure 3.

The design of the MDCNN-VGG fully considers the different background environments of different underwater images, and considers the situation of different domains (e.g., different shooting perspectives, different light intensities, etc.). Different channels input different domains of underwater image data, which enhances the model domain adaptation capability through the parameter-sharing mechanism [32], and designs the DCNN to effectively extract the underwater image features.

The objective function of our model is $L_{e}$ for a newly designed externally supervised VGG-based loss function, in addition to $L_{\mathrm{cl}}$ and $L_{MSE}$, and is defined as
(8)$$L_{e}=\frac{1}{n}\sum_{c}{\left(A^{c}-H^{c}\right)}^{2}$$
where $H^{c}$ denotes additional multi-domain supervision, for example, the multi-domain segmentation mask for Figure 1. Owing to the high time complexity of generating pixel-level segmentation maps, the model designed in this study was expected to use a small amount of data under external supervision to meet the requirements of practical scenarios.

The VGG was introduced into the model, and all parameters were shared between the two network streams and the VGG. The error values of the enhanced and real images were passed to the VGG to obtain the corresponding feature representations; that is, the distance between the two types of images, *I* and *I**, was calculated based on perceptual loss.

Naturally, the final MDCNN-VGG loss function, *L*, can be obtained using Equation (9).
(9)$$L=L_{\mathrm{cl}}+\alpha L_{MSE}+\beta L_{e}$$
where $L_{\mathrm{cl}}$ and $L_{MSE}$ are shown in Section 3.2; α,β are coefficients that control the degree of involvement of the DCNN and additional supervision in the MDCNN-VGG training. The MDCNN-VGG can be easily improved for other vision tasks [33,34]. When the final network output $f_{l,k}$ is obtained, $L_{e}$ is used to direct the network model to the key regions of the task of interest.

## 4. Experimental Design and Result Analysis

The main contents of the experiments in this section include the dataset and experimental setup, qualitative evaluation, quantitative evaluation, ablation experiment, multi-domain scenario, visual perception effect experiment, and inference time complexity.

### 4.1. Dataset and Experimental Setup 

#### 4.1.1. Dataset

We tested the MDCNN-VGG on real image datasets to demonstrate its ability to enhance underwater images from different datasets. The datasets used can be considered multi-domain underwater images, and the specific datasets are described as follows:

UFO-120 [43]: Clear images were collected from ocean soundings for different water types. The corresponding underwater images were generated using style transformation, where a subset of 120 images was used as the test set.

EUVP Dark [13]: A large collection of 10 K paired and 25 K unpaired images collected by data producers during ocean soundings under various visibility conditions, with both poor and good perceptual quality. It contains 5500 pairs of images with dark underwater backgrounds. In this study, 1000 images were used to test the model.

UIEBD [18]: Comprises 890 pairs of underwater images captured under different lighting conditions with different color ranges and contrasts.

#### 4.1.2. Experimental Configuration

The model in this study was trained using the Adam optimizer, with the learning rate set to 0.0001, dropout set to 0.5, and batch size set to 1000. Approximately 10 h was required to optimize the model with more than 50 cycles. The experiment was run on an Intel(R) Core (TM) i7-107000k CPU with a 16 GB RAM and an NVIDIA GTX 1080 GPU.

#### 4.1.3. Baseline 

Shallow UWnet [4]: A shallow CNN for underwater image enhancement. Three methods, white balance, gamma correction, and histogram equalization, were used to preprocess the WaterNet enhancement input for the characteristics of the blurred background environment of underwater images.

UResnet [9]: A CNN-based residual network is a more comprehensive supervised learning method for underwater image enhancement.

FUnIE GAN [10]: Assumes nonlinear mapping between distorted and enhanced images, and the blurring of images is removed by constructing image enhancement.

CycleGAN [12]: A technique that automatically performs image-to-image transformation without pairwise examples, using a batch of images from the source and target domains that do not need to be correlated and trained in an unsupervised manner.

UGAN-P [34]: Underwater GAN with gradient penalty.

Uw HL [44]: Color recovery based on fuzzy lines. This method was based on a physical model design scheme.

#### 4.1.4. Evaluation Metrics 

The standard metrics used in this study, namely the peak signal-to-noise ratio (PSNR) and structural similarity index metric (SSIM), were quantitatively evaluated for the output images of the proposed model. The PSNR and SSIM quantify the reconstruction quality and structural similarity of the output images with regard to the corresponding reference images [45]. In addition, the output image quality was analyzed in this study using the non-reference underwater image quality metric (UIQM). The UIQM is composed of three attribute metrics: image color (UICM), sharpness (UISM), and contrast (UIConM), where each attribute assesses the quality of the underwater image from a single dimension.

### 4.2. Multi-Domain Scenarios 

In this study, the image enhancement effect of the model was verified through multi-domain underwater image enhancement experiments. The enhancement effects of the different models and algorithms for different domain images are shown in Figure 4 and Figure 5, respectively.

From Figure 4, it is clear that the different methods have some enhancement effect, but poor performance for different domain images. As shown in Figure 4, the Shallow UWnet caused overexposure, whereas the FUnIE GAN deepened the background color of the water body. However, the physics-based scheme, Uw-HL, exhibits image oversaturation, which makes the contrast of underwater images in different domains too high. The reason for the failure of these schemes is that they ignore the correlation between underwater images in different domains. The proposed model has the best enhancement effect for images in different domains, where the targets are still clearly visible, even for images in different viewpoints. The success of the model in this study can be attributed to the designed multichannel and fusion of different loss terms that are used to achieve a better enhancement effect for underwater images from different viewpoints. In other scenarios, such as those shown in Figure 5, the enhancement performance of the model in this study remained the best.

Figure 6 shows qualitative comparisons of the different underwater image enhancement schemes.

As shown in Figure 6, FUnIE GAN and CycleGAN often appear oversaturated, whereas UResnet and Shallow UWnet usually fail to correct the green tones in the images, owing to the greater depth of the above network models and the tendency of the model to overfit. UGAN-P and Uw-HL performed better and their enhancement performances were similar to those of the MDCNN-VGG. However, the UGAN-P and Uw-HL models are susceptible to the influence of bright objects in the scene, and the oversaturation phenomenon, particularly Uw-HL, fails to enhance the global brightness in some cases, which shows that the above two schemes cannot explicitly pre-estimate the targets in underwater images and fail to improve the visual perception of the images. However, it can be observed from Figure 6 that the multichannel setup of the MDCNN-VGG achieves color consistency and hue correction, and enhances the reference color or texture information in the loss function using the multichannel fusion processing of underwater images from different domains. Overall, the multi-domain underwater image enhancement scheme achieves the same performance as the physical-based model without using scene depth or unknown water body information and outperforms other baselines [46].

### 4.3. Qualitative Evaluation

Table 1 presents a comparison of the average PSNR, structural similarity (SSIM), and UIQM of the test images for each model. The results demonstrate that the MDCNN-VGG achieves the best UIQM values in UFO-120; however, the PSNR and SSIM values are 0.30 relatively.

Weak in the paired EUVP dataset. The UGAN-P and UResNet produced better results for the paired data. Similar analyses were performed in [38,39], which quantified the sharpness, clarity, and contrast of underwater images. The UIQM results presented in Table 1 show that the MDCNN-VGG outperforms state-of-the-art methods, and the best UIQM values show that the resulting image has balanced color, clarity, and contrast. In this paper, we hypothesize that the global similarity loss in the MDCNN-VGG and FUnIE GAN, or the gradient penalty term in UGAN-P contributes to such enhancement tasks, owing to the fact that all the above methods add L1 terms to the adversarial objective. It is evident from Table 1 that the MDCNN-VGG contributes to an average improvement over the state-of-the-art Shallow Uwnet in the UIQM metric for different datasets by 0.30, where comparable statistics of performance improvement are observed for the PSNR and SSIM [47,48].

CNNs have a wide range of applications in computer vision, and owing to their advantages, they have been promoted for applications in underwater imaging. The MDCNN-VGG maintained excellent quantitative performance. The enhancement capability of the MDCNN-VGG for underwater images in different datasets is shown in Figure 7, Figure 8 and Figure 9. Notably, the MDCNN-VGG can test its generalization capability on different datasets, making it more widely applicable to various types of underwater scenes with different degradation levels.

### 4.4. Ablation Experiments 

In this study, we first qualitatively analyzed the enhanced the color and sharpness of images generated with the MDCNN-VGG and compared them with their respective corresponding baselines. As shown in Figure 10, the enhanced underwater images mainly recover their true colors and sharpness. In addition, the color correction and global contrast enhancement results plotted for underwater images of different hues, as shown in Figure 11, clearly show the distinct texture and vivid colors of the local images after underwater image enhancement with the proposed model. This is due to the additional multi-domain supervised VGG setup that allows for the pixel-level segmentation masking of underwater images, using Le to guide the network model to focus on the critical tasks of interest regions [49].

To verify the contribution of each loss term to the underwater image enhancement (see Figure 12), we have looked at their effect on the image enhancement with and without these loss terms. Figure 12 shows that the different loss terms clearly contribute to the image.

Enhancement can be analyzed more intuitively via a direct comparison using the UIQM values. The calculation shows that in the top row of Figure 12, the UIQM value in Figure 12c is higher than the UIQM result of Figure 12b by 0.038, whereas the UIQM result in Figure 12d is higher than that of Figure 12b by 0.321. The UIQM result of the model utilizing all loss terms is higher than that of Figure 12b by 0.546. This clearly shows that the different loss terms designed in this study can result in better image enhancement and compensate for each other’s deficiencies. The image in the bottom row of Figure 12 shows the effect of the above analysis.

## 5. Conclusions

In this study, we propose a deep learning model, the MDCNN-VGG, which is an underwater image enhancement technique based on a hybrid model, to achieve the fast enhancement of multi-domain underwater images. In the MDCNN-VGG, a DCNN using different channels can effectively mine the local information of underwater images in different domains and pass the above local information to the VGG to enhance underwater images accurately. In this study, the model was based on the global color and structural content of the image, the local texture, and style information, and the perceived loss function was established by evaluating the image quality. We performed extensive qualitative and quantitative evaluations as well as multi-domain image enhancement studies. The results show that the MDCNN-VGG contributes a 0.11 average improvement over the state-of-the-art Shallow UWnet for different datasets in terms of UIQM values, and other performance metrics are similarly improved.

In the future, we believe that (1) we can improve the design structure of the model to further enhance its application inference capability, such as considering image pair applications for underwater images in a small sample drive [47]; (2) even though the model in this study enhances image texture and color enhancement capability through different domains of underwater images collaboratively, there are still problems such as blurred details, color bias, and overexposure. For the different loss terms in this study, robust optimization through adversarial learning was used to enhance the model’s ability to deal with these detailed problems and further suppress the perturbation of samples to the model through robust optimization.

## Figures and Tables

**Figure 1 sensors-23-08983-f001:**
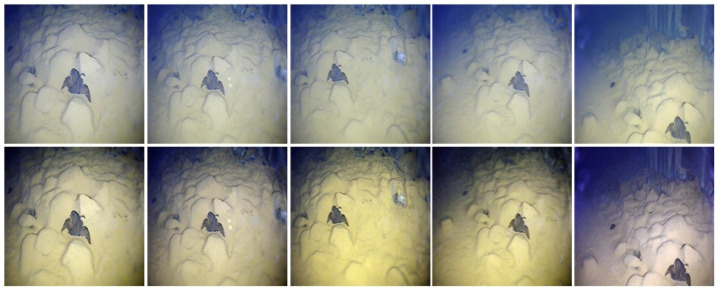
Multi-domain underwater image enhancement.

**Figure 2 sensors-23-08983-f002:**
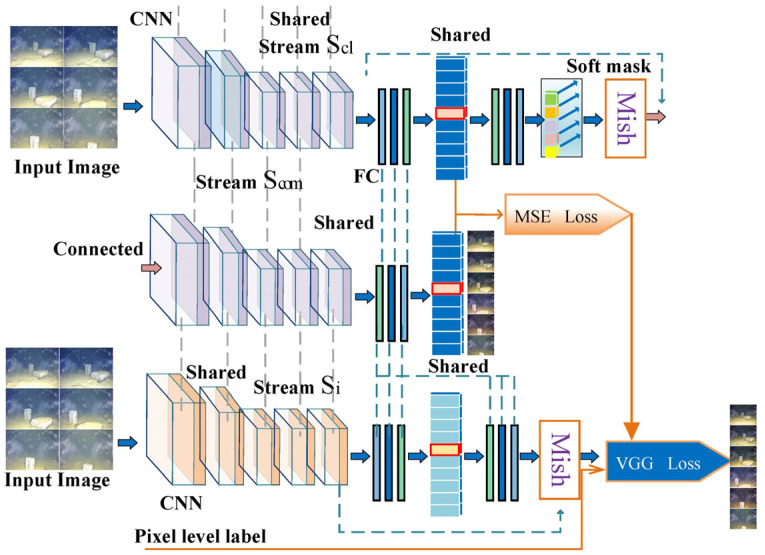
MDCNN-VGG architecture.

**Figure 3 sensors-23-08983-f003:**
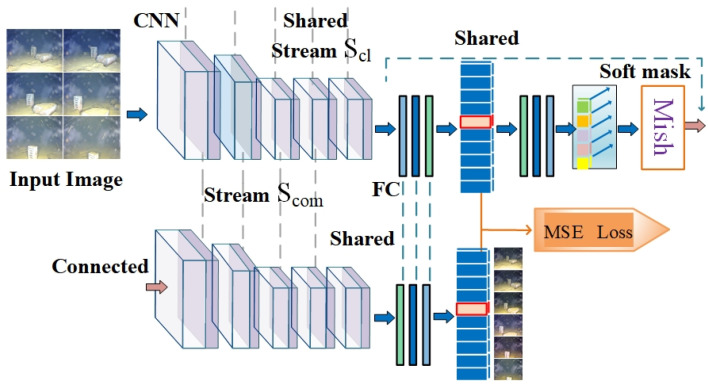
Single-channel DCNN framework diagram.

**Figure 4 sensors-23-08983-f004:**
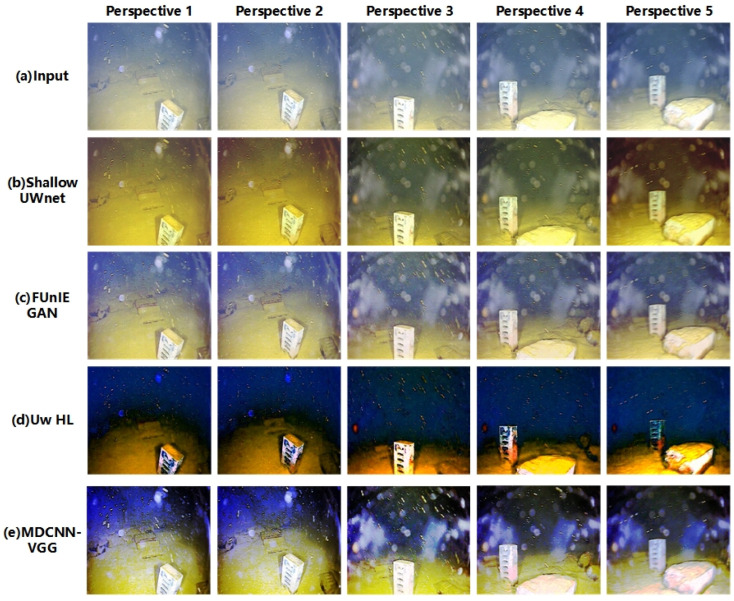
Submarine target scene.

**Figure 5 sensors-23-08983-f005:**
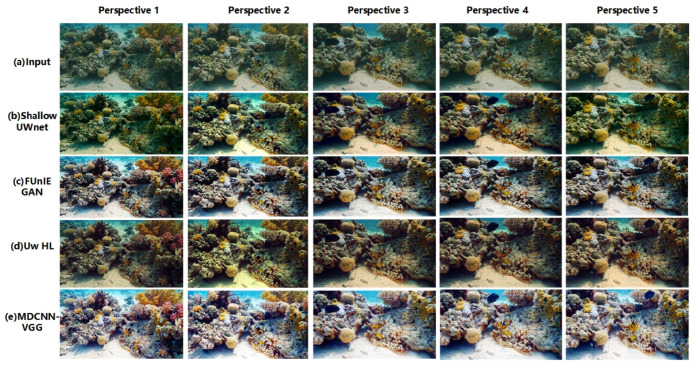
Submarine coral group scene.

**Figure 6 sensors-23-08983-f006:**
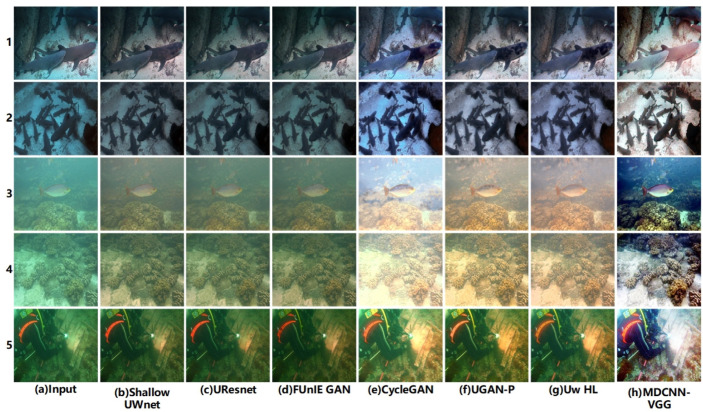
Qualitative comparisons of different underwater image enhancement schemes.

**Figure 7 sensors-23-08983-f007:**
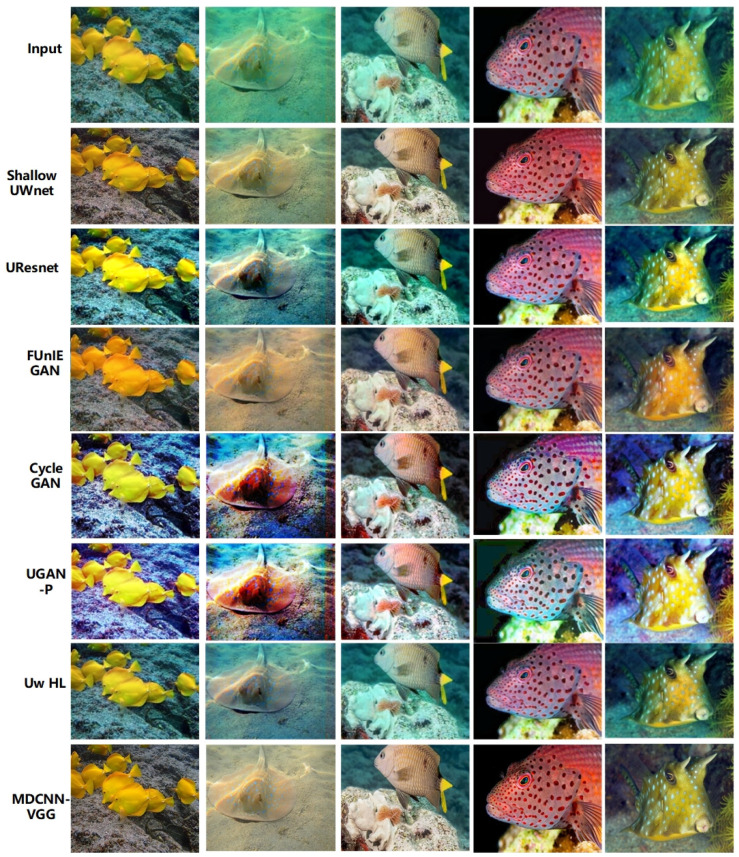
Underwater image enhancement results of UFO-120.

**Figure 8 sensors-23-08983-f008:**
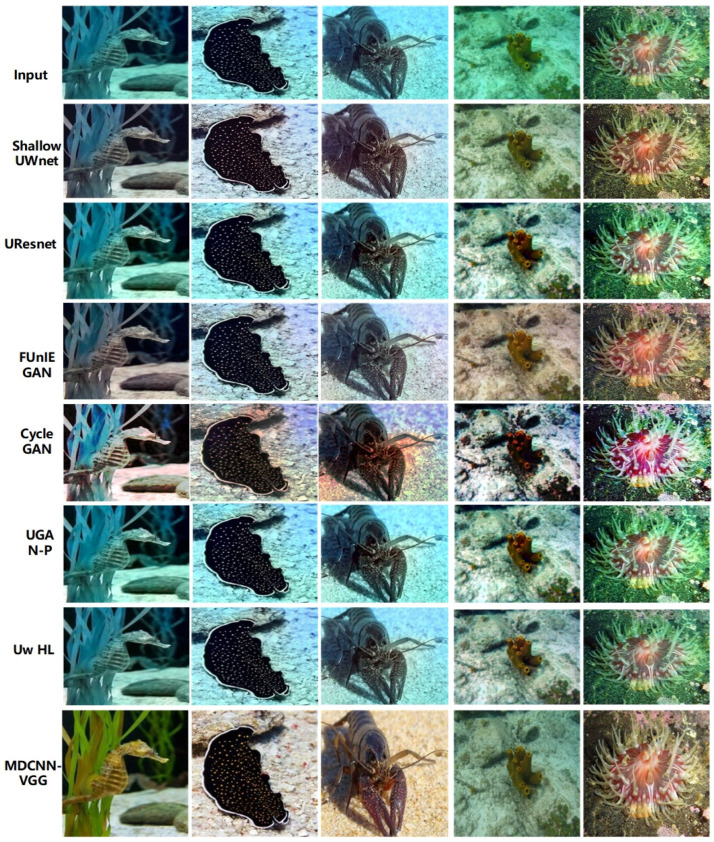
Underwater image enhancement results of EUVP.

**Figure 9 sensors-23-08983-f009:**
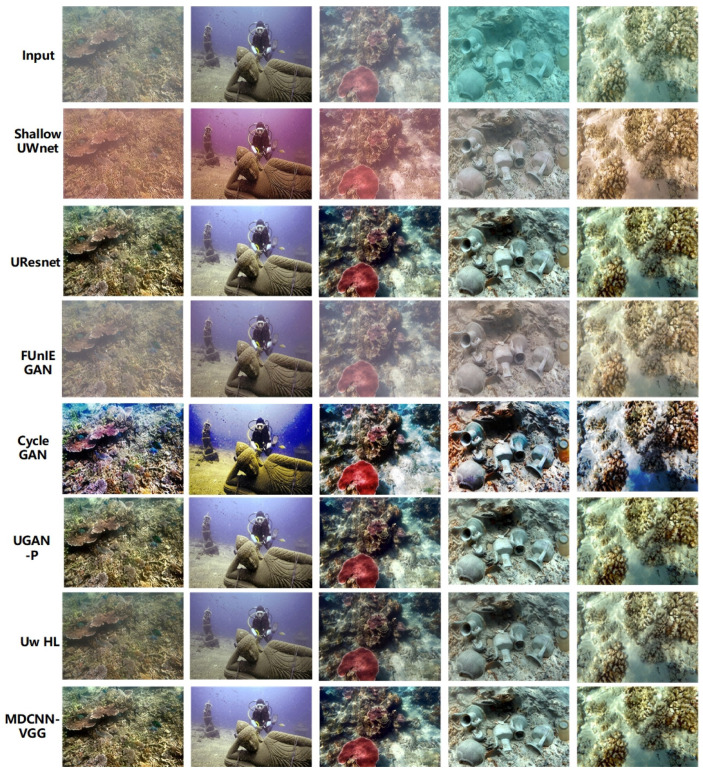
Underwater image enhancement results of UIEB.

**Figure 10 sensors-23-08983-f010:**
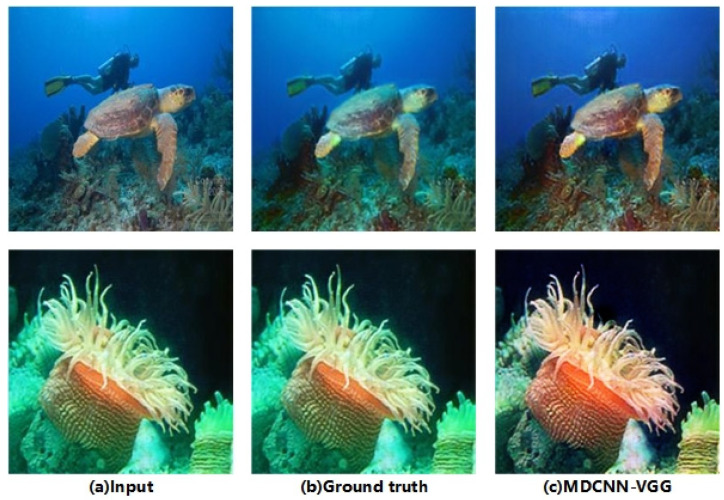
Underwater image enhancement results.

**Figure 11 sensors-23-08983-f011:**
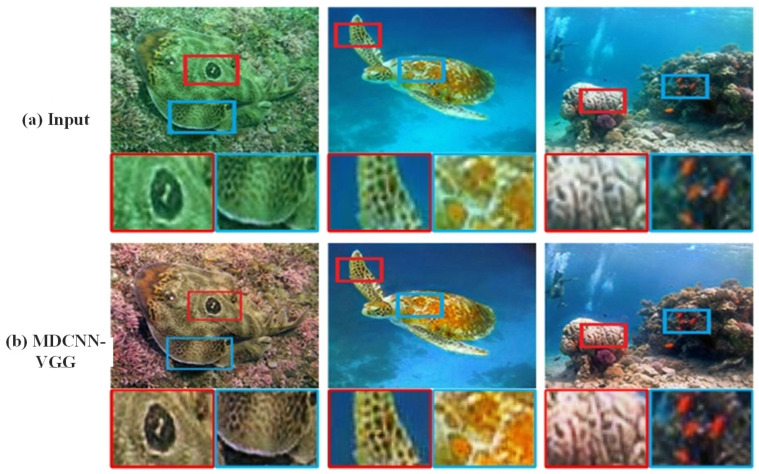
Underwater image detail enhancement results.

**Figure 12 sensors-23-08983-f012:**
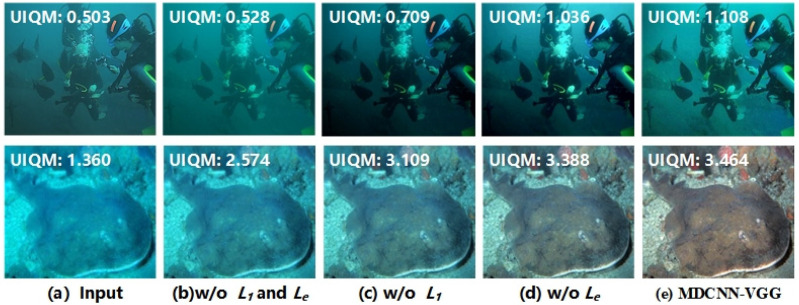
Ablation experiment results of different loss items in the MDCNN-VGG.

**Table 1 sensors-23-08983-t001:** Quantitative metrics of underwater image enhancement.

	PSNR			SSIM			UIQM		
	EUVP Dark	UFO-120	UIEB	EUVP Dark	UFO-120	UIEB	EUVP Dark	UFO-120	UIEB
Shallow UWnet	20.83	18.45	21.24	0.90	0.73	0.90	2.71	2.56	2.50
UResnet	27.61	21.24	24.98	0.97	0.78	0.95	2.40	2.27	2.38
FUnIE GAN	28.68	30.38	38.75	0.96	0.81	1.00	2.95	2.89	3.08
CycleGAN	8.79	16.23	17.24	0.84	0.68	0.79	2.95	2.89	2.77
UGAN-P	27.61	15.23	24.96	0.97	0.67	0.95	2.40	2.73	2.38
Uw HL	39.91	30.38	38.75	1.00	0.81	0.99	2.71	2.56	2.50
MDCNN-VGG	27.49	25.27	19.09	0.82	0.74	0.75	3.00	2.88	2.80

## Data Availability

The authors confirm that the data supporting the findings of this study are available from the corresponding author, upon reasonable request.

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
