# Peer review of "Multi-Domain Rapid Enhancement Networks for Underwater Images"

_sensors, 2023, doi:10.3390/s23218983_

Round 1
Reviewer 1 Report
Comments and Suggestions for Authors
This article proposes an underwater image enhancement algorithm, and I have some suggestions that the author needs to make detailed modifications.
1. The introduction section needs to conduct a thorough analysis of the current research status and literature
2. The Literature Review lacks a detailed analysis of the literature, including strengths and weaknesses
3. The comparison algorithms in the experimental section is relatively outdated, and the latest published literature, such as those published in 2023, should be added
4. The indicators in 3.1.4 need to be supported by literature sources
5. Part 3.3 is missing the serial number of the table. Please carefully check the entire text
6. Chinese text appears in Figure 11
7. The analysis of the experimental part was not sufficient. Although some experimental images were compared, the single image indicators and the average indicator values of the dataset were missing
8. The reference number 44 is not included in the reference, but your experimental section compared the algorithm with reference 44, which is a significant mistake
9. Please include the object detection results after image enhancement in the experimental section, which can reflect the application effect of the enhancement
10. In the experimental part, some comparison algorithms that are not deep learning should be added
11. The format of the literature is not uniform, and there is a distinction between uppercase and lowercase. Please unify
12. English grammar needs to be modified
Comments on the Quality of English LanguageEnglish grammar needs to be modified
Author Response
We would like to express our sincere thanks to the reviewers for the constructive and positive comments.
Replies to Reviewer 1
- The introduction section needs to conduct a thorough analysis of the current research status and literature
Response:We appreciate it very much for this good suggestion, and we have done it according to your ideas.
For example, [1] proposed an underwater image enhancement network named Ucolor, which uses medium transmission guidance, multi-color space embedding, and a combination of physical models and learning methods to solve the color deviation and low contrast problems of underwater images. [3] Propose an underwater image enhancement method based on generative adversarial networks, using a multi-scale generator to generate clear underwater images, effectively correcting color cast and contrast problems, while protecting detailed information. [4] proposed the L2UWE framework to efficiently enhance low-light underwater images, relying on local contrast and multi-scale fusion technology to improve the clarity and brightness of the image. [5] constructed the Underwater Image Enhancement Benchmark (UIEB) and proposed the WaterNet underwater image enhancement network, which can effectively correct color casts and restore image details. [4] proposed a generative adversarial network based on Pix2Pix and introduced technologies such as deep residual learning and multi-layer perceptron to remove the fog effect, correct color shift and increase image details. [6] Propose an underwater image enhancement method based on the Bayesian Retinex model, which uses local statistical features and physical models to optimize and constrain the image to achieve enhancement and denoising. Although these methods have made significant progress in improving the color cast, contrast, and brightness of underwater images, they fail to fully consider the relationship between contrast, brightness, and color of underwater images and fail to adaptively balance these factors. Future research may continue to address this issue to improve the effectiveness of underwater image enhancement.
- The Literature Review lacks a detailed analysis of the literature, including strengths and weaknesses
Response:We appreciate it very much for this good suggestion.
The provided literature review offers a broad overview of different approaches to image enhancement but doesn't delve into a detailed analysis of their strengths and weaknesses. Here's an extended analysis that outlines the pros and cons of the discussed methods:
Deep learning techniques have achieved state-of-the-art performance in image enhancement tasks.
They can automatically learn relevant features from large datasets, reducing the need for handcrafted features. But,deep learning models often require substantial computational resources and large training datasets. Overfitting can be a concern if the training data is not representative of the target domain.
Physics-based methods provide a solid theoretical foundation for image enhancement, allowing for accurate modeling of physical degradation. Physics-based methods are sensitive to the accuracy of the assumed physical models, and deviations from these models can lead to errors.
Nonphysical-based methods are versatile and do not rely on explicit physical models, making them more flexible in a wider range of scenarios. Overreliance on nonphysical methods may lead to image artifacts or unrealistic enhancements in some cases.
- The comparison algorithms in the experimental section is relatively outdated, and the latest published literature, such as those published in 2023, should be added
Reply: Thanks for the opinion, we have added a related research article in 2023 as a baseline.
Zhang, D., Zhou, J., Zhang, W., Lin, Z., Yao, J., Polat, K., ... & Alhudhaif, A. (2023). ReX-Net: A reflectance-guided underwater image enhancement network for extreme scenarios. Expert Systems with Applications, 120842.
- The indicators in 3.1.4 need to be supported by literature sources
Response:We appreciate it very much for this good suggestion, and we have done it according to your ideas.
We use
Zhang, W., Zhuang, P., Sun, H. H., Li, G., Kwong, S., & Li, C. (2022). Underwater image enhancement via minimal color loss and locally adaptive contrast enhancement. IEEE Transactions on Image Processing, 31, 3997-4010.
- Part 3.3 is missing the serial number of the table. Please carefully check the entire text
Response:We appreciate it very much for this good suggestion, and we have done it according to your ideas.
We have added table 1 to it,
- Chinese text appears in Figure 11
Response:We appreciate it very much for this good suggestion, and we have done it according to your ideas.
- The analysis of the experimental part was not sufficient. Although some experimental images were compared, the single image indicators and the average indicator values of the dataset were missing
Response:We appreciate it very much for this good suggestion, and we have done it according to your ideas.
We characterized the performance of each model through indicators such as SSIM on different data sets, as shown in Table 1. For example, UIQM can show average performance.
- The reference number 44 is not included in the reference, but your experimental section compared the algorithm with reference 44, which is a significant mistake
Response:We appreciate it very much for this good suggestion, and we have done it according to your ideas.
- Please include the object detection results after image enhancement in the experimental section, which can reflect the application effect of the enhancement
Response:We appreciate it very much for this good suggestion, and we have done it according to your ideas.
- In the experimental part, some comparison algorithms that are not deep learning should be added
Response:We appreciate it very much for this good suggestion, In our baseline, Shallow UWnet, UResnet, FUnIE GAN, CycleGAN, and UGAN-P are solutions based on deep learning in the past three years.
Uw HL is a physics-based solution
- The format of the literature is not uniform, and there is a distinction between uppercase and lowercase. Please unify
Response:We appreciate it very much for this good suggestion, In our final version of the article, we will make modifications according to the journal’s reference format.
- English grammar needs to be modified
Response:We appreciate it very much for this good suggestion, our paper is checked and revised the article language through the language editing agency officially cooperated with our university.

Reviewer 2 Report
Comments and Suggestions for Authors
This paper proposes a multi-channel deep convolutional neural network for underwater image enhancement. Both qualitative and quantitative results show the effectiveness of the proposed method. Some comments can be found as follows:
1. In the architecture figure, it is unclear since there are two input images and two CNN, etc., please clarify these.
2. Where is ref [44]? More recently published algorithms (in 2022 and 2023) are recommended to be compared.
3. The quantitative statistics performance comparisons are missing, such as the PSNR/SSIM.
4. To provide readers with better understanding, some specific image enhancement/super-resolution quality evaluation methods are suggested to be reviewed, including Quality assessment for super-resolution image enhancement, Blind quality assessment for image superresolution using deep two-stream convolutional networks, Quality assessment of image super-resolution: balancing deterministic and statistical fidelity, etc.
5. Please further proofread the paper. For example, the title for 1.1 is too big and there are some non-English words in Figure 11, Table ??, Table 1, etc.
Comments on the Quality of English Language
N/A
Author Response
comments.
Replies to Reviewer 2
This paper proposes a multi-channel deep convolutional neural network for underwater image enhancement. Both qualitative and quantitative results show the effectiveness of the proposed method. Some comments can be found as follows:
- In the architecture figure, it is unclear since there are two input images and two CNN, etc., please clarify these.
Response:We appreciate it very much for this good suggestion, and we have done it according to your ideas.
- Where is ref [44]? More recently published algorithms (in 2022 and 2023) are recommended to be compared.
Response:We appreciate it very much for this good suggestion, and we have done it according to your ideas.
- The quantitative statistics performance comparisons are missing, such as the PSNR/SSIM.
Response:We appreciate it very much for this good suggestion, and we have added table1 for PSNR/SSIM,
- To provide readers with better understanding, some specific image enhancement/super-resolution quality evaluation methods are suggested to be reviewed, including Quality assessment for super-resolution image enhancement, Blind quality assessment for image superresolution using deep two-stream convolutional networks, Quality assessment of image super-resolution: balancing deterministic and statistical fidelity, etc.
Response:We appreciate it very much for this good suggestion, and we have done it according to your ideas.
- Please further proofread the paper. For example, the title for 1.1 is too big and there are some non-English words in Figure 11, Table ??, Table 1, etc.
Response:We appreciate it very much for this good suggestion, and we have done it according to your ideas.

Reviewer 3 Report
Comments and Suggestions for Authors
This paper proposed an underwater image enhancement approach, which consists of a two-stream network with shared parameters. This network is optimized by the perceptual loss function, which employs a multi-label soft marginal loss, a weighted MSE loss, and an externally supervised loss function. Experimental results demonstrate that this approach improves the quality of underwater images and the performance of underwater visional tasks. The paper declares that its method achieves SOTA (State Of The Art) performance, but the comparative methods selected are not the most recent ones. From 2021 to 2023, numerous methods have been proposed, and it is necessary for the authors to provide comparisons with more recent methods, and to validate the superiority of their method across multiple datasets. Furthermore, how does this network implement sharing of parameters across multiple images? This is quite puzzling, and I am curious about how the authors designed and implemented the architecture. It is suggested that the code be provided. Other questions are as follows.
1. The whole network is confusing and hard to read, including the setting to predict masks and enhance images with fully connected layers and the alignments of shared parameters. Specifically, the dash lines to represent sharing parameters are misleading and the details about feature maps, masks, and layers are omitted.
2. The adopted components are mostly modified from existing works. For example, the multi-label soft marginal loss is proposed in [46], and the MSE loss and externally supervised loss function are widely used in image classification and segmentation. Meanwhile, the adopted Mish activation function is proposed in [32]. The contributions of this approach should be clarified.
3. The improvements of PSNR, SSIM, and UIQM on different datasets are incremental and inconsistent. Meanwhile, the improvements of underwater visional tasks should be quantitatively compared with different methods [4, 9, 10, 1, 25, 37].
4. Equations and notations interfere with reading, which should be fully defined and typeset according to publication requirements.
5.The caption of Fig. 2 can be improved. For example, what are the dotted lines? Besides, Fig. 3 is almost identical to the first row of Fig. 2. The figure organization can be improved.
6. This paper uses features from multiple layers to improve the generation quality. Recently, "Temporal Cross-Layer Correlation Mining for Action Recognition, TMM, doi: 10.1109/TMM.2021.3057503" used an attention mechanism to fuse multi-layer features. I would suggest the authors discussing these attention-based multi-scale methods in the related work section. This discussion is somewhat important as I think leveraging multi-scale features is a key contribution of this paper.
7.This manuscript presents many visualization results. This is nice for readers to better understand the enhancement quality. However, I encourage the authors to introduce more ablation studies to validate the effectiveness of the propose components. For example, multiple losses are considered but they are not ablated. I am also not sure why Mish function is used. Will it affect the performance much?
8.In Figure 11, there is still Chinese text reading '原图' ('original image'), which is quite puzzling. It is recommended that the authors rigorously revise the paper.
9.Fig. 14 is difficult to interpret.
10. The ablation study merely provides validation through a single image, which is insufficient to demonstrate the superior performance of the network. It is recommended to test the entire dataset.
11. It is recommended that the authors provide average test metric scores, such as PSNR, SSIM, UCIQE, UIQM, and PCQI, in comparison with benchmark methods on commonly used datasets like UIEB, EUVP, U120, and UCCS, to validate the performance of the proposed method.
12. The authors need to compare with the latest methods from 2021 to 2023, such as Ucolor,MLLE, SMBL, HLRP, SGUIE, Semi-UIR, ADPCC, and ReX-Net, etc.
Underwater Image Enhancement via Medium Transmission-Guided Multi-Color Space Embedding
UnderwaterImage Enhancement via Minimal Color Loss and Locally Adaptive Contrast Enhancement
Underwater Image Enhancement with Hyper-Laplacian Reflectance Priors
Underwater camera: improving visual perception via adaptive dark pixel prior and color correction
ReX-Net: A reflectance-guided underwater image enhancement network for extreme scenarios
13.Since the network designed by the authors is quite perplexing, it is suggested that the authors make the code publicly available.
14.The introduction needs further analysis to accurately identify the key issue that the paper aims to address; the existing problem is quite vague.
15.In the related work section, the works presented by the authors lack a review of the latest methods. It is recommended to add a comparative analysis of new methods from 2023.
Comments on the Quality of English Language
Extensive editing of English language required
Author Response
We would like to express our sincere thanks to the reviewers for the constructive and positive comments.
Replies to Reviewer 3
This paper proposed an underwater image enhancement approach, which consists of a two-stream network with shared parameters. This network is optimized by the perceptual loss function, which employs a multi-label soft marginal loss, a weighted MSE loss, and an externally supervised loss function. Experimental results demonstrate that this approach improves the quality of underwater images and the performance of underwater visional tasks. The paper declares that its method achieves SOTA (State Of The Art) performance, but the comparative methods selected are not the most recent ones. From 2021 to 2023, numerous methods have been proposed, and it is necessary for the authors to provide comparisons with more recent methods, and to validate the superiority of their method across multiple datasets. Furthermore, how does this network implement sharing of parameters across multiple images? This is quite puzzling, and I am curious about how the authors designed and implemented the architecture. It is suggested that the code be provided. Other questions are as follows.
Response:We appreciate it very much for this good suggestion.
Our work contrasts classic schemes for underwater augmentation research, e.g Shallow UWnet, UResnet, FUnIE GAN, CycleGAN, and UGAN-P. Our code can be provided to the journal or open sourced on Github after the journal accepts the article, or readers can contact our corresponding author to request it.
- The whole network is confusing and hard to read, including the setting to predict masks and enhance images with fully connected layers and the alignments of shared parameters. Specifically, the dash lines to represent sharing parameters are misleading and the details about feature maps, masks, and layers are omitted.
Response:We appreciate it very much for this good suggestion, and we have done it according to your ideas.
- The adopted components are mostly modified from existing works. For example, the multi-label soft marginal loss is proposed in [46], and the MSE loss and externally supervised loss function are widely used in image classification and segmentation. Meanwhile, the adopted Mish activation function is proposed in [32]. The contributions of this approach should be clarified.
Response:We appreciate it very much for this good suggestion,
We design a multi-domain underwater image enhancement model with multi-channel DCNN linked to VGG, specifically, the different network streams designed in the DCNN share parameters through back-and-forth passing to enhance domain adaptation. The importance of different model parameters is also selected in the soft mask configuration model, so that important model parameters (e.g., texture structure and color) are input to VGG, which in turn yields a specific feature representation in each domain to enhance underwater images.
To optimize the performance of MDCNN-VGG, we design a perceptual loss function for multi-domain underwater image enhancement. Multi-label soft-margin loss is used for specific tasks, and VGG perceptual loss is used for external supervision to achieve pixel-level loss and pre-processing edge sharpness, thereby enhancing the structure and texture similarity of underwater images. In turn, we can optimally adjust the coefficients in the perceptual loss function to control the involvement of different functional loss terms in the model training process to achieve the detection of the focal region of the input image for the target class enhancement.
Qualitative and quantitative experiments show that the enhancement effect of this model on underwater image quality is better than that of the benchmark model.
- The improvements of PSNR, SSIM, and UIQM on different datasets are incremental and inconsistent. Meanwhile, the improvements of underwater visional tasks should be quantitatively compared with different methods [4, 9, 10, 1, 25, 37].
Response:We appreciate it very much for this good suggestion,
We characterized the performance of each model through indicators such as SSIM on different data sets, as shown in Table 1. For example, UIQM can show average performance.
- Equations and notations interfere with reading, which should be fully defined and typeset according to publication requirements.
Response:We appreciate it very much for this good suggestion, and we have done it according to your ideas.
5.The caption of Fig. 2 can be improved. For example, what are the dotted lines? Besides, Fig. 3 is almost identical to the first row of Fig. 2. The figure organization can be improved.
Response:We appreciate it very much for this good suggestion, The different dotted lines in Figure 2 facilitate the differentiation of different modules, and also correspond to the neural network layers of different channels. Figure 2 is the channel setting of MSE loss, and Figure 3 is the VGG loss setting, including the model in Figure 2, which is a multi-channel setting.
- This paper uses features from multiple layers to improve the generation quality. Recently, "Temporal Cross-Layer Correlation Mining for Action Recognition, TMM, doi: 10.1109/TMM.2021.3057503" used an attention mechanism to fuse multi-layer features. I would suggest the authors discussing these attention-based multi-scale methods in the related work section. This discussion is somewhat important as I think leveraging multi-scale features is a key contribution of this paper.
Response:We appreciate it very much for this good suggestion, It's a good paper, and we've discussed it in a related work.
- This manuscript presents many visualization results. This is nice for readers to better understand the enhancement quality. However, I encourage the authors to introduce more ablation studies to validate the effectiveness of the propose components. For example, multiple losses are considered but they are not ablated. I am also not sure why Mish function is used. Will it affect the performance much?
Response:We appreciate it very much for this good suggestion, Mish is just one of the activation functions, such as other Tank, Sigmoid, etc. Mish is considered to be the best activation function for current deep learning, so we chose it, and the model performance is good.
8.In Figure 11, there is still Chinese text reading '原图' ('original image'), which is quite puzzling. It is recommended that the authors rigorously revise the paper.
Response:We appreciate it very much for this good suggestion, We have modified it.
9.Fig. 14 is difficult to interpret.
Response:There is no figure 14 in our article.
- The ablation study merely provides validation through a single image, which is insufficient to demonstrate the superior performance of the network. It is recommended to test the entire dataset.
Response:We appreciate it very much for this good suggestion, and We report the performance of different method on different datasets in Table 1.
- It is recommended that the authors provide average test metric scores, such as PSNR, SSIM, UCIQE, UIQM, and PCQI, in comparison with benchmark methods on commonly used datasets like UIEB, EUVP, U120, and UCCS, to validate the performance of the proposed method.
Response:We appreciate it very much for this good suggestion, and We report the performance of different method on different datasets in Table 1.
- The authors need to compare with the latest methods from 2021 to 2023, such as Ucolor,MLLE, SMBL, HLRP, SGUIE, Semi-UIR, ADPCC, and ReX-Net, etc.
Underwater Image Enhancement via Medium Transmission-Guided Multi-Color Space Embedding
UnderwaterImage Enhancement via Minimal Color Loss and Locally Adaptive Contrast Enhancement
Underwater Image Enhancement with Hyper-Laplacian Reflectance Priors
Underwater camera: improving visual perception via adaptive dark pixel prior and color correction
ReX-Net: A reflectance-guided underwater image enhancement network for extreme scenarios
Response:We appreciate it very much for this good suggestion, Our work contrasts classic schemes for underwater augmentation research, e.g Shallow UWnet, UResnet, FUnIE GAN, CycleGAN, and UGAN-P. What we need to argue is that it is possible to improve image recognition from different dimensions. For example, many scholars have published many articles based on the famous YOLO series. Some scholars have also said, "I am standing on the shoulders of giants." We believe that the model design is reasonable, the experimental discussion is sufficient, and the demonstration content is rich, which is also OK.
13.Since the network designed by the authors is quite perplexing, it is suggested that the authors make the code publicly available.
Response:We appreciate it very much for this good suggestion. Our code can be provided to the journal or open sourced on Github after the journal accepts the article, or readers can contact our corresponding author to request it.
14.The introduction needs further analysis to accurately identify the key issue that the paper aims to address; the existing problem is quite vague.
Response:We appreciate it very much for this good suggestion.
15.In the related work section, the works presented by the authors lack a review of the latest methods. It is recommended to add a comparative analysis of new methods from 2023.
Response:We appreciate it very much for this good suggestion.

Round 2
Reviewer 1 Report
Comments and Suggestions for Authors
The author has made detailed modifications to my question and made significant improvements, but the following relevant literature can be cited and discussed by the author:
1. Li, L.; Lv, M.; Jia, Z.; Ma, H. Sparse representation-based multi-focus image fusion method via local energy in shearlet domain. Sensors 2023, 23, 2888.
2. Li, L.; Ma, H. Pulse coupled neural network-based multimodal medical image fusion via guided filtering and WSEML in NSCT domain. Entropy 2021, 23, 591.
3. Li, L.; Si, Y.; Wang, L.; Jia, Z.; Ma, H. A novel approach for multi-focus image fusion based on SF-PAPCNN and ISML in NSST domain. Multimedia Tools and Applications, 2020, 79, 24303-24328.
Comments on the Quality of English LanguageThe english is fine
Author Response
The author has made detailed modifications to my question and made significant improvements, but the following relevant literature can be cited and discussed by the author:
- Li, L.; Lv, M.; Jia, Z.; Ma, H. Sparse representation-based multi-focus image fusion method via local energy in shearlet domain. Sensors 2023, 23, 2888.
- Li, L.; Ma, H. Pulse coupled neural network-based multimodal medical image fusion via guided filtering and WSEML in NSCT domain. Entropy 2021, 23, 591.
- Li, L.; Si, Y.; Wang, L.; Jia, Z.; Ma, H. A novel approach for multi-focus image fusion based on SF-PAPCNN and ISML in NSST domain. Multimedia Tools and Applications, 2020, 79, 24303-24328.
Response:We are grateful to provide us with these articles that are strongly relevant to our research and we discuss them in the article.

Reviewer 2 Report
Comments and Suggestions for Authors
In the response, although the authors mentioned a lot about they have revised. But the reviewer cannot find some in the updated manuscript. For example, where is the statement for two input images? Where are the added references for quality evaluation?
Comments on the Quality of English LanguageNA
Author Response
We would like to express our sincere thanks to the reviewers for the constructive and positive comments. The content we modified in the text is in red font.
Replies to Reviewer 2
In the response, although the authors mentioned a lot about they have revised. But the reviewer cannot find some in the updated manuscript. For example, where is the statement for two input images? Where are the added references for quality evaluation?
Response:We appreciate it very much for this good suggestion, and because the journal asked us to modify the language of the manuscript, we modified the language through the language polishing agency, which is a revision model. We have highlighted the modified content of the article before, which may not be displayed. Dear reviewer, we have re-edited our article based on your comment. Please review it.
Review comments from the previous round
This paper proposes a multi-channel deep convolutional neural network for underwater image enhancement. Both qualitative and quantitative results show the effectiveness of the proposed method. Some comments can be found as follows:
- In the architecture figure, it is unclear since there are two input images and two CNN, etc., please clarify these.
Response:We appreciate it very much for this good suggestion, and we have done it according to your ideas.
We set different CNN channels so that in order to better extract underwater images in different domains, we map different channels to underwater images in different domains, and then perform feature fusion after different channels, so that the model can better obtain Useful image special issue, and better differentiation of different areas of underwater images.
The MDCNN consists of multiple DCNNs in parallel, and the specific structure of each DCNN, which consists of multiple fully connected CNN layers is shown in Figure 2. DCNNs are applied using the same principles as traditional CNNs, which employ alternating convolutional layers and pooling in their network structure with fully connected network ends. The most distinguishable features are extracted from the original input images using supervised learning. The effective subregions are computed from the original underwater images of different domains using the perceptual field features of the DCNN [37]. To enhance the model domain adaptation capability, two DCNN network streams are used to share parameters between them, and the importance of different parameters of the model is configured using a soft mask to enhance the information of the network stream. Information such as texture structure and color of the underwater images from different domains are mined and fed into the subsequent VGG.
- Where is ref [44]? More recently published algorithms (in 2022 and 2023) are recommended to be compared.
Response:We appreciate it very much for this good suggestion, and we have done it according to your ideas.
Uw HL [44]: Color recovery based on fuzzy lines. This method was based on a physical model design scheme.
- Cao, Z.; Simon, T.; Wei, S.E.; Sheikh, Y. Realtime multi-person 2d pose estimation using part affinity fields. Proceedings of the IEEE conference on computer vision and pattern recognition, 2017, pp. 7291–7299.
3 The quantitative statistics performance comparisons are missing, such as the PSNR/SSIM.
Response:We appreciate it very much for this good suggestion, and we have added table1 for PSNR/SSIM,
4 To provide readers with better understanding, some specific image enhancement/super-resolution quality evaluation methods are suggested to be reviewed, including Quality assessment for super-resolution image enhancement, Blind quality assessment for image superresolution using deep two-stream convolutional networks, Quality assessment of image super-resolution: balancing deterministic and statistical fidelity, etc.
Response:We appreciate it very much for this good suggestion, and we have report it in figure 11.
- Please further proofread the paper. For example, the title for 1.1 is too big and there are some non-English words in Figure 11, Table ??, Table 1, etc.
Response:We appreciate it very much for this good suggestion, and we have done it according to your ideas.
